# Accuracy and Safety Between Robot-Assisted and Conventional Freehand Fluoroscope-Assisted Placement of Pedicle Screws in Thoracolumbar Spine: Meta-Analysis

**DOI:** 10.3390/medicina61040690

**Published:** 2025-04-09

**Authors:** Alberto Morello, Stefano Colonna, Enrico Lo Bue, Giulia Chiari, Giada Mai, Alessandro Pesaresi, Diego Garbossa, Fabio Cofano

**Affiliations:** 1Neurosurgery Unit, Department of Neuroscience “Rita Levi Montalcini”, AOU Città della Salute e della Scienza di Torino, University Hospital, University of Turin, 10126 Turin, Italy; stefano.colonna@unito.it (S.C.); enrico.lobue@unito.it (E.L.B.); giada.mai@unito.it (G.M.); alessandro.pesaresi@unito.it (A.P.); diego.garbossa@unito.it (D.G.); fabio.cofano@unito.it (F.C.); 2BCAM Bilbao Center for Applied Mathematics—Mazarredo Zumarkalea, 48009 Bilbao, Bizkaia, Spain; gchiari@bca-math.org

**Keywords:** spine surgery, arthrodesis, fluoroscopy, pedicle screw, robotic surgery, screw placement

## Abstract

*Background and Objectives*: Robotic-assisted surgery (RS) has progressively emerged as a promising technology in modern thoracolumbar spinal surgery, offering the potential to enhance accuracy and improve clinical outcomes. To date, the benefits of robot-assisted techniques in thoracolumbar spinal surgery remain controversial. The objective of this study was to assess the efficacy and safety of RS compared to fluoroscopy-assisted surgery (FS) in spinal fusion procedures. *Materials and Methods*: In accordance with the PRISMA guidelines, a systematic review and meta-analysis was conducted, using REVMAN V5.3 software. The review protocol was registered in the Prospective Register of Systematic Reviews (PROSPERO) website with the following registration number: CRD42024567193. *Results*: Eighteen studies were included in the meta-analysis with a total of 1566 patients examined. The results demonstrated a worse accuracy in FS in cases with major violations of the peduncular cortex (D–E grades, according to Gertzbein’s classification) [(odds ratio (OR) 0.47, 95%-CI 0.28 to 0.80, I^2^ 0%]. In addition, a lower complication rate was shown in the RS group compared to the FS group, specifically regarding the need for surgical revision due to screw mispositioning (OR 0.28-CI 0.17 to 0.48, I^2^ 98%). *Conclusions*: Advantages of robot-assisted techniques were demonstrated in terms of postoperative complications, revision surgery rates, and the accuracy of screw placement. While RS represents a valuable and promising technological advancement in thoracolumbar spinal surgery, future studies are needed to further explore its advantages in thoracolumbar spinal surgery and to identify which spinal surgical approach has greater advantages when using the robot.

## 1. Introduction

Spinal pathologies represent a significant public health problem and could derive from different etiologies; degenerative conditions represent the most common cause due to progressive aging of the population [1]. Regardless of its etiology, the pedicle screw fixation technique remains the most widespread method for spinal arthrodesis. The traditional placement of pedicle screws is with the freehand technique, which includes manual execution, surgical experience to understand the anatomy, and the use of intraoperative fluoroscopy [2,3,4]. However, the traditional freehand screw placement method has several limitations, specifically regarding its accuracy leading to potential pedicle violations [5]. As reported in the literature, the rate of misplacement for conventional freehand pedicle screws ranges from 5% to 41% in the lumbar spine and from 3% to 55% in the thoracic spine. This misplacement can cause potential significant complications, such as nerve injury or dural tearing [5,6]. The field of spinal surgery has undergone significant transformations in recent decades to address this challenge. Robot-assisted pedicle screw placement has gradually emerged as an alternative to freehand techniques, demonstrating advantages in terms of improved accuracy and reduced intraoperative bleeding [7]. Several recent meta-analyses have tried to define the role of robot-assisted surgery in terms of screw accuracy, suggesting that this technique could improve short-term clinical outcomes and shorten recovery times [8,9]. Nonetheless, some aspects remain unclear, and various studies show non-conclusive results when considering the application of robot-assisted techniques for thoracolumbar fusion surgery. Moreover, given their high acquisition and maintenance costs, it remains uncertain whether the use of robots justifies their integration into routine hospital practice [10,11]. The purpose of this meta-analysis is to evaluate the role of robotic-assisted surgery (RS) with a holistic perspective, including more parameters compared to previously available studies and considering both lumbar and thoracic screws, aiming to better define its potential superiority compared to freehand techniques.

## 2. Materials and Methods

### 2.1. Literature Search

This systematic review was conducted in accordance with the PRISMA protocol (Preferred Reporting Items for Systematic Reviews and Meta–Analyses) [12]. Relevant studies were sourced from PubMed/MEDLINE, Embase, and the Cochrane Library. The final search was performed on 20 June 2024. The following terms were used for the database: (“Robot assisted” OR “Robotic” OR “Robotic Surgery”) AND (“Fluoroscopy-guided” OR “Fluoroscopy” [Mesh]) AND (“Freehand”) AND (“Pedicle screw” OR “Spinal Fusion” [Mesh] OR “Spinal Diseases/surgery” [Mesh]). Additionally, to ensure a thorough review, the reference lists of all relevant articles were manually checked to identify any other studies of potential significance.

### 2.2. Inclusion and Exclusion Criteria

Comparative studies in English that met the following PICO (Patient, Intervention, Comparison, Outcome) criteria were deemed eligible. Patients: patients of any age and sex with any pathology of the thoracolumbar spine. Intervention: robot-assisted surgery (RS) for the placement of pedicle screws in spinal surgeries. Comparison: freehand fluoroscope-assisted surgery (FS). Main Outcome: accuracy of pedicle screw position. Study design: randomized controlled trials (RCTs), prospective studies (not necessarily randomized), and retrospective cohort studies (RCSs).

The inclusion criteria were as follows: comparative studies between RS and FS treatments, articles with postoperative computed tomography scans to assess accuracy, and articles providing sufficient data for meaningful comparison (more than 10 pedicle screws per study group). The exclusion criteria were as follows: duplicate publications; articles lacking a control group with traditional freehand pedicle screw placement; meta-analyses, case reports, review articles, or studies with fewer than 5 patients per group; cadaver studies; as well as laboratory and animal studies.

### 2.3. Protocol Registration

The review protocol was registered and published in Prospective Register of Systematic Reviews (PROSPERO) (www.crd.york.ac.uk/PROSPERO, accessed on 20 July 2024) website with registration number: CRD42024567193.

### 2.4. Screening and Full-Text Review

Title and abstract screening, full-text review, and data extraction were conducted concurrently by two reviewers (G.M. and E.L.B.). Any disagreements at any stage were resolved through discussion and consensus. In cases of unresolved disagreements, a third reviewer (A.M.) was involved to reach a resolution. The process was facilitated using Rayyan—version number 1.6.0 [13].

### 2.5. Data Extraction

The essential data extracted from eligible studies included the first author, publication date, study type, patient sample size, screw sample size, age, gender distribution, type of robot used, surgical segment, follow-up duration, and the accuracy of pedicle screw placement. The other outcomes extracted were as follows: surgical parameters (operation time, intraoperative blood loss, and superior facet joint violation (FJV)), clinical indexes [Visual Analog Scale (VAS), Oswestry Disability Index (ODI), hospital stays, complications including durotomy, poor healing, infections, and necessary surgical revisions], and total radiation dosage.

The accuracy of pedicle screw position was evaluated using the Gertzbein scale, which categorizes screw position into five grades, with grade A representing the highest precision and grade E representing the lowest precision [14]. The superior FJV was assessed according to Babu et al. [15].

### 2.6. Risk-of-Bias Assessment

The Cochrane risk-of-bias tool for nonrandomized studies of interventions (ROBINS-I tool—version number 2) was used to assess the risk of bias in the included studies [16]. This was performed by two authors (S. C. and A. M.).

### 2.7. Statistical Analysis

The statistical analysis was performed using Python (release Python 3.10.11). The baseline characteristics of the included studies were analyzed using descriptive statistics. The meta-analysis was performed using the online tool https://metaanalysisonline.com/ (accessed on 10 January 2025). A random effects model was employed for the meta-analysis due to the methodological and clinical heterogeneity among the studies. The odds ratio (OR) for frequency data, along with the corresponding 95% confidence intervals (CI), were pooled using the inverse-variance test. A combined analysis of DerSimonian–Laird and Paul–Mandel estimators was applied to account for between-study heterogeneity, which refers to the proportion of total variation attributable to differences among the included studies, rather than sampling error. All statistical tests were two tailed, and the significance level was set at *p* value < 0.05.

## 3. Results

### 3.1. Literature Search

As shown in the PRISMA flowchart of Figure 1, the PubMed/MEDLINE, Embase, and Cochrane Library search yielded 2881 articles. After removing duplicates, 2143 records were screened, and 631 underwent a full-text assessment for eligibility. Finally, 18 studies that met the inclusion and exclusion criteria and reported on 1566 patients were identified and included in the quantitative synthesis. Sufficient data were available to perform the meta-analysis for blood loss, hospitalization days, operation time, radiation dose, accuracy of pedicle screw position, FJV, VAS, ODI, and complications.

### 3.2. Study Characteristics and Quality

Appendix A lists the main characteristics of the included studies, including the publication year; study design; sample size; vertebral level; operation time; blood loss; accuracy of pedicle screw position; FJV, VAS, and ODI difference between pre- and postoperative; total complications; and the risk of bias evaluated with the ROBINS-I tool. These 18 studies yielded 1566 patients. A total of 788 underwent FS and 778 underwent RS. The mean age ranged from 14.49 to 72.3 years (FS: 14.49–68.0 years; RS: 14.69–72.3 years). The diagnoses were varied: spinal stenosis, lumbar spondylolisthesis, and lumbar disk herniation. Only one study included spinal fractures. The total number of screws placed ranged from 40 to 172 in FS and from 72 to 158 in RS. The single segment (two adjacent vertebrae) was the most common arthrodesis in both FS and RS. Among the 18 studies, the type of robot most used was the Renaissance. The total complications ranged from 0 to 22 cases in FS and from 0 to 15. Poor healing, including infections, was the most common complication in 23 FS cases and 15 RS cases, respectively.

Ten studies were randomized controlled trials. The prospective and retrospective study designs were used in studies 1 and 7, respectively. All included studies demonstrated a satisfactory quality, according to the Cochrane’s ROBINS-I tool (Appendix A).

### 3.3. Quantitative Results (Figure 2)

▪
**Screw accuracy of pedicle screw position (A)**


Altogether, 13 studies were analyzed with a total of 2467 subjects in the RS cohort and 2283 subjects in the FS cohort. There is a statistically significant difference between the two cohorts, the overall OR is 2.43 with a 95% confidence interval of 1.49–3.87. The test for the overall effect shows a significance at *p* < 0.05.
▪**Screw accuracy of pedicle screw position (B)**

Altogether, 13 studies were analyzed with a total of 2467 subjects in the RS cohort and 2283 subjects in the FS cohort.

There is a statistically significant difference between the two cohorts; the overall OR is 0.48 with a 95% confidence interval of 0.32–0.71. The test for the overall effect shows a significance at *p* < 0.05.
▪**Screw accuracy of pedicle screw position (C)**

Altogether, 12 studies were analyzed with a total of 2467 subjects in the RS cohort and 2283 subjects in the FS cohort.

There is no statistically significant difference between the two cohorts; the overall OR is 0.54 with a 95% confidence interval of 0.28–1.06. The test for the overall effect does not show a significant effect.
▪**Screw accuracy of pedicle screw position (D–E)**

Altogether, six studies were analyzed with a total of 2467 subjects in the RS cohort and 2283 subjects in the FS cohort.

There is a statistical difference between the two cohorts; the overall OR is 0.47 with a 95% confidence interval of 0.28–0.8. The test for the overall effect shows a significance at *p* < 0.05.
▪**Intraoperative blood loss**

Altogether, eight studies were analyzed with a total of 390 subjects in the RS cohort and 388 subjects in the FS cohort. There is no statistically significant difference between the two cohorts; the summarized mean difference (MD) is −11.89 with a 95% confidence interval of −167.33–143.56. The test for the overall effect does not show a significant effect.
▪**Hospital stays**

Altogether, 10 studies were analyzed with a total of 486 subjects in the RS cohort and 433 subjects in the FS cohort. There is no statistical difference between the two cohorts; the summarized MD is −0.73 with a 95% confidence interval of −1.82–0.36. The test for the overall effect does not show a significant effect.
▪**Operation time**

Altogether, 14 studies were analyzed with a total of 628 subjects in the RS cohort and 590 subjects in the FS cohort. There is no statistical difference between the two cohorts; the summarized MD is 27.32 with a 95% confidence interval of −12.35–66.98. The test for the overall effect does not show a significant effect.
▪**Radiation dose exposure**

Altogether, four studies were analyzed with a total of 219 subjects in the RS cohort and 215 subjects in the FS cohort. There is no statistical difference between the two cohorts; the summarized MD is −41.05 with a 95% confidence interval of −115.33–33.23. The test for the overall effect does not show a significant effect.
▪**Superior facet joint violation**

Altogether, two studies were analyzed with a total of 432 subjects in the RS cohort and 454 subjects in the FS cohort.

There is a statistical difference between the two cohorts with regard to grades 1 and 2, according to Babu et al. [15]. The test for the overall effect shows a significance at *p* < 0.05.
▪**Total complications**

Altogether, 12 studies were analyzed with a total of 638 subjects in the RS cohort and 703 subjects in the FS cohort.

There is a statistically significant difference between the two cohorts; the overall OR is 0.28 with a 95% confidence interval of 0.17–0.48. The test for the overall effect shows a significance at *p* < 0.05. 

**Figure 2 medicina-61-00690-f002:**
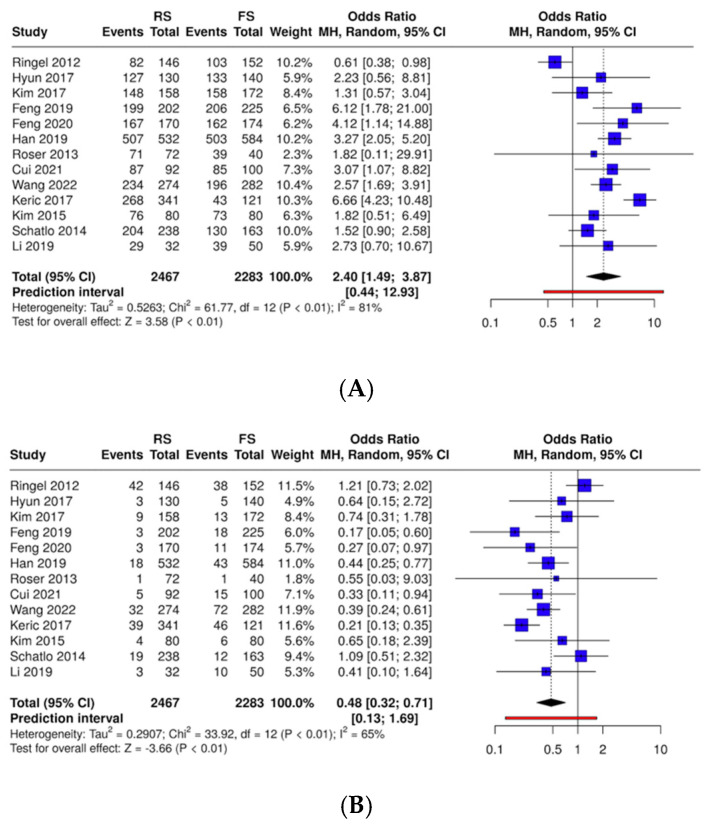
Forest plots of the pooled analysis of the investigated outcome variables [11,17,18,19,20,21,22,23,24,25,26,27,28,29,30,31,32]. RS: robotic-assisted surgery; FS: fluoroscopy-assisted surgery. (**A**) Accuracy of screw placement (Grade A); (**B**) accuracy of screw placement (Grade B); (**C**) accuracy of screw placement (Grade C); (**D**) accuracy of screw placement (Grades D and E); (**E**) intraoperative blood loss; (**F**) hospital stays; (**G**) operation time; (**H**) radiation dose exposure; (**I**) superior facet joint violation (grade 0); (**J**) superior facet joint violation (grade 1); (**K**) superior facet joint violation (grade 2); (**L**) superior facet joint violation (grade 3); and (**M**) total complications.

### 3.4. Publication Bias

Funnel plots of the accuracy of the pedicle screw position can be found in Figure 3. The funnel plot does not indicate a potential publication bias. The Egger’s test does not support the presence of funnel plot asymmetry (*p* > 0.05). Funnel plots for the other outcomes investigated are available in the Appendix A.

## 4. Discussion

RS has progressively emerged as a promising technology in modern thoracolumbar spinal surgery, offering the potential to enhance accuracy and to improve clinical outcomes. The integration of imaging guidance, real-time navigation, and robotic arms aim to enhance surgical precision and facilitate minimally invasive techniques to improve postoperative pain and recovery. Since its introduction in surgical practice, abundant scientific literature has been published regarding the advantages and disadvantages of robot-assisted thoracolumbar spinal fusion surgery, with heterogeneous and often contradictory results [33]. To date, the benefits of robot-assisted techniques in thoracolumbar spinal surgery remain controversial. In this context, this study aims to explore the benefits and drawbacks of robot-assisted thoracolumbar spinal fusion, based on evidence from the relevant scientific literature, and offer a holistic perspective on its role in modern spinal surgery.

This paper provides a complete and holistic meta-analysis identifying the differences between RS and FS in thoracolumbar spinal fusion surgery. Overall, 18 studies were included in the meta-analysis with a total of 1566 patients examined. The results showed no statistically significant differences between the RS and FS groups regarding the intraoperative blood loss, length of hospitalization, radiation exposure, and operative time. On the other hand, the results showed a worse accuracy in the FS in cases with major violations of the peduncular cortex (grades D and E). Lastly, we demonstrated a minor complication rate in the RS group compared to the FS group, specifically regarding the need for surgical revision due to screw mispositioning.

Several studies have investigated whether the use of robot-assisted techniques guarantees minor intraoperative blood loss. A prospective randomized controlled trial by Han et al. in 2019 demonstrated a significant reduction in intraoperative blood loss in the RS group compared to the FS group [11]. Similar results were obtained from previous meta-analyses and comparative studies available in the literature [5,34]. Conversely, in a recent prospective comparative study, Wang and colleagues excluded significant differences between RS and FS in terms of intraoperative blood loss [24]. In this context, our results from the analysis of 778 patients from eight different studies revealed the absence of significant advantages of the RS group compared to the FS group in terms of intraoperative blood loss.

The length of hospitalization (LoH) is a critical parameter in the evaluation of post-surgical recovery. Shorter LoHs typically reflect effective postoperative care and can indicate the quality of the surgical technique and perioperative care. In addition, shorter LoHs have a fundamental cost-effectivene implication in terms of healthcare resource allocation. To date, the literature offers heterogeneous evidence on whether the application of RS techniques can impact the LoH. Despite studies available in the literature demonstrating some advantages of RS techniques in the reduction in LoH [18,19,24], our results on a total of 919 patients did not demonstrate statistically significant differences between RS and FS in terms of hospitalization duration, in accordance with previous high-quality studies [8,11,24,26].

Since the introduction of robotic-assisted techniques in spinal surgery, operative time (OT) has been one of the most thoroughly explored aspects of its application. Even though the evidence from the literature is often contradictory, most of the studies demonstrated the absence of clear advantages of RS in the reduction in OT compared to fluoroscopy techniques [5,8,9,11,22,26]. Only a limited number of studies demonstrated an improvement of OT with robotic-assisted techniques [34,35]. In this context, however, the test for the overall effect does not show a significant effect, our results demonstrated higher OTs in patients treated with RS compared to FS. These data can be explained by the fact that robotic operating systems are generally complex, requiring longer operative times during the surgeon’s learning curve.

Despite the abundance of studies addressing radiation exposure during RS and FS, evidence from the literature is still inconclusive. Results from high-quality studies have demonstrated the advantages of robotic-assisted techniques in terms of radiation exposure compared to FS [11,22,25,34,35]. Nonetheless, in contrast with these results, several articles have highlighted the absence of clear improvements in radiation exposure with RS [9,24,26,36]. In accordance with the latter studies, the results of our meta-analysis on 434 patients did not demonstrate a significant reduction in radiation exposure with RS techniques when compared to FS.

Historically, the accuracy of the screw placement represents one of the most analyzed aspects of spinal instrumentation. Despite the lack of a unanimous consensus, several articles from the literature have demonstrated a higher accuracy and precision of screw placement with RS compared to FS [9,11,19,20,21,22,23,27,28,29]. Nonetheless, some studies have revealed comparable results in terms of the screw accuracy between robotic- and fluoroscopy-assisted techniques [8,19,20,21,22,23,24,25,26,27,28,29,30,31,32]. In addition, a few studies have demonstrated the higher accuracy of FS to RS [26]. With regard to screw placement in terms of the absence of pedicle cortex violations (grade A according to Gertzbein’s classification), the results of our meta-analysis of 4750 screws shows a higher accuracy in FS cases. On the other hand, the results showed a worse accuracy in the FS in cases with major violations of the peduncular cortex (grades D and E).

A FJV during thoracolumbar spinal fusion surgery can lead to significant complications, including pain, biomechanical instability, delayed fusion, and nerve damage. In addition, FJVs can increase the risk of adjacent segment degeneration. These factors can negatively affect postoperative recovery, extend the LoH, increase the likelihood of revision surgery, and ultimately worsen long-term outcomes. Several articles have demonstrated lower FJV rates after robotic procedures compared to fluoroscopy techniques [24,31,35,37]. Nonetheless, our results from 886 patients demonstrated a statistical difference between RS and FS in terms of FJVs grade 1 and 2.

Several studies available in the literature have investigated whether the application of robotic techniques to thoracolumbar spinal fusion surgery could be beneficial in terms of postoperative complications. Most of these studies did not find any significant differences between RS and FS in terms of peri- and postoperative complications [8,9,32,34]. Nevertheless, some authors have demonstrated more favorable postoperative complication in patients treated with robotic techniques [23,37]. The results of this meta-analysis supported the evidence of a lower rate of postoperative complications (including durotomy, poor healing, infections, and necessary surgical revisions) in the robotic cohort compared to FS. These results were consistent with our findings in terms of screw accuracy, more specifically with the higher Gertzbein gradings of mispositioned screws in the FS group compared to robotic cohort, which could consequently justify a higher rate of revision surgery in the group treated with FS techniques.

The typology of robotic systems is highly diverse. The robots utilized in this analysis are as follows: “TiRobot system” (TINAVI Medical Technologies Co., Ltd., Beijing, China), “SpineAssistTM” (Mazor Robotics, Caesarea, Israel), “Renaissance” (Mazor Robotics Ltd., Caesarea, Israel), “ExcelsiusGPS” (Globus Medical), “TianJi Robot System”, and the “Orthbot system.” Other major systems approved by the FDA include “ROSA” (Zimmer-Biomet), “Cirq” (Brainlab), “Curexo” (CUREXO Inc), “Kinguide” (Point Robotics), and “Remi” (Accelus). The cost of each platform ranges from USD 700,000 to USD 1.5 million. In a meta-analysis comparing the performance of the three most popular robotic platforms from 2016 to 2023, Maclean et al. reported excellent accuracy rates among the various robots (Mazor, Rosa, and ExcelsiusGPS) [38]. On the other hand, they reported that both the ExcelsiusGPS and Mazor were associated with significantly reduced blood loss compared to ROSA. According to the data from our analysis, the accuracy appears to be high across all the described robots. Specifically, the “TiRobot system” and “SpineAssistTM” appear to achieve optimal precision in terms of pedicle screw position accuracy. However, due to various confounders, such as the heterogeneity of the underlying diseases and the number of patients involved, definitive conclusions regarding which robotic system is superior cannot yet be drawn.

In addition, it is important to analyze the usefulness of the robot in the performance of different surgical approaches to the spine in the future. For instance, the use of the robot may have a significant impact on cortical bone trajectory (CBT) screws, as the CBT technique increases the contact surface area between the screws and cortical bone compared to the traditional method [39,40]. According to the planned trajectory, the robotic arm moves to the targeted entry point. Once in position, the cannulas and dilators can be placed through the end effector. The use of robots may be particularly useful for CBT screws, as this technique follows a divergent trajectory, reducing the need for skeletonization. Therefore, the maneuvering angle of the robotic arm could facilitate this technique. In fact, in some cases, for classical pedicle screws using robots, it is necessary to create a trans-fascial cut through the same open wound or a separate percutaneous passage down to the bone. However, additional studies are required to determine the actual benefit of the robot in the CBT spinal technique.

In conclusion, while RS in spinal thoracolumbar fusion offers potential benefits, the results of this meta-analysis suggest that the advantages do not automatically translate into significantly better clinical outcomes when compared to traditional freehand techniques. The robotic approach often entails higher costs, longer operative times, and a steep learning curve, which may offset its theoretical benefits. Moreover, the improvements in accuracy and safety, though promising, are not universally observed across all studies. Therefore, while robot-assisted techniques represent an exciting technological advancement, their routine adoption should be guided by a critical evaluation of patient-specific factors, surgeon expertise, and institutional resources. On the other hand, the results showed a worse accuracy in cases with major violations of the peduncular cortex (grades D and E) and a higher rate of postoperative complications (including durotomy, poor healing, infections, and necessary surgical revisions) in the FS group compared to robotic cohort.

This study has limitations to be considered. A significant factor lies in the high heterogeneity of the results from the articles considered for the meta-analysis, despite the overall quality of evidence being considered as high. The studies included in the meta-analysis consider different surgical approaches and techniques, which introduces variability in outcomes and complicates the comparison of results. Furthermore, the use of different types of robotic systems and technologies across the studies adds complexity to the analysis, as each robot may have distinct capabilities and limitations that influence the surgical performance and outcomes. Lastly, the evaluation of screw accuracy varies among the studies, with differences in measurement methods and criteria. These discrepancies underline the challenges of drawing definitive conclusions from the existing literature and highlight the need for more standardized protocols and long-term studies to provide clearer guidance on the effectiveness of robot-assisted spinal surgery.

## 5. Conclusions

The results of this meta-analysis demonstrated the absence of significant advantages of RS when applied to thoracolumbar fusion in terms of the intraoperative blood loss, length of hospitalization, radiation exposure, and operative time. Nonetheless, the advantages of RS were demonstrated in terms of postoperative complications, revision surgery rates, and the accuracy of screw placement. While spinal robotic surgery represents a valuable and promising technological advancement in thoracolumbar spinal surgery, future studies are needed to further explore its advantages in thoracolumbar spinal surgery. In addition, in the future it is important to identify which spinal surgical approach has greater advantages when using the robot.

## Figures and Tables

**Figure 1 medicina-61-00690-f001:**
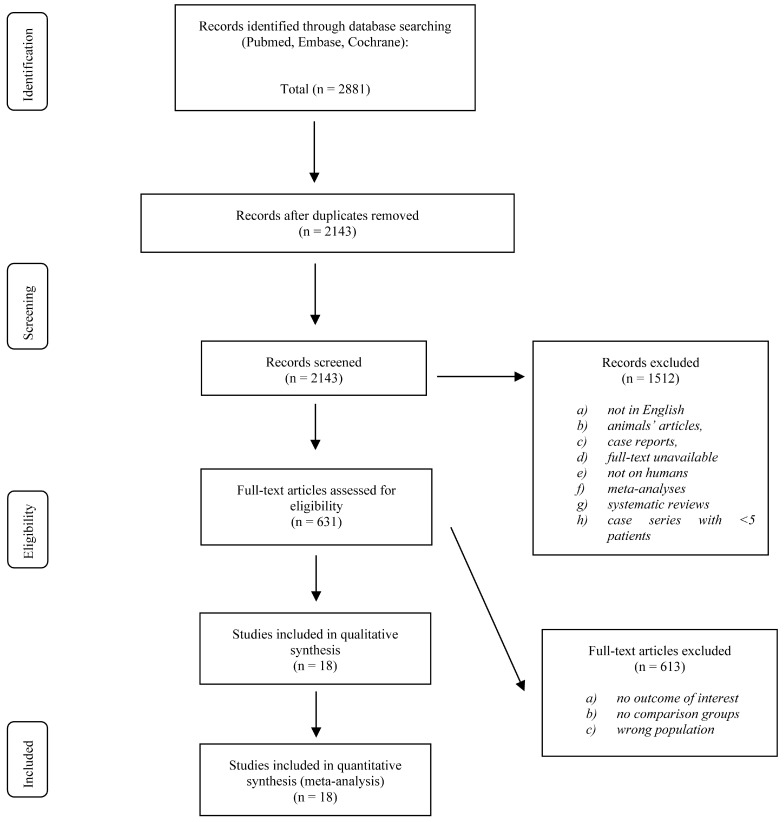
PRISMA flowchart.

**Figure 3 medicina-61-00690-f003:**
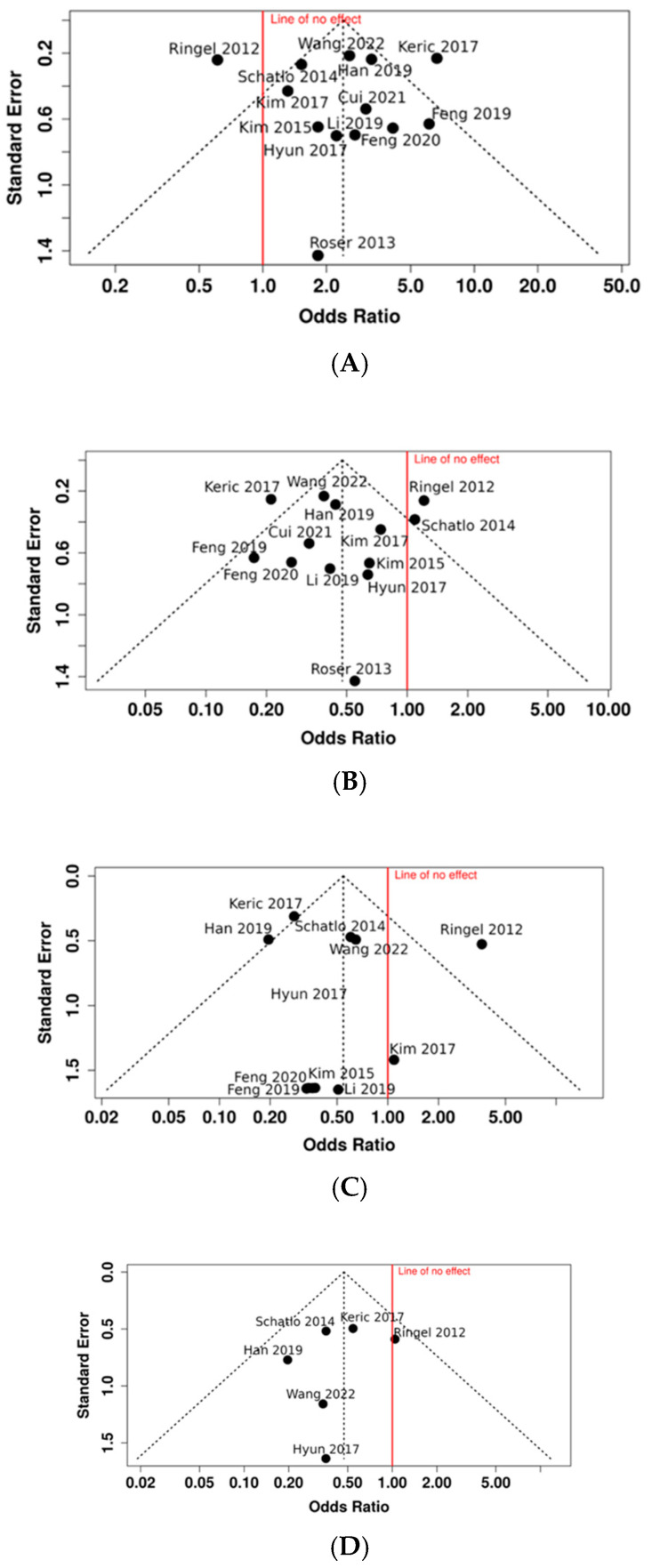
Funnel plots [11,18,19,20,22,23,24,25,26,28,29,30,31]. (**A**) Publication bias on accuracy of screw placement (Grade A); (**B**) publication bias on accuracy of screw placement (Grade B); (**C**) publication bias on accuracy of screw placement (Grade C); and (**D**) publication bias on accuracy of screw placement (Grades D and E).

## Data Availability

Any data or information needed to reproduce the findings presented are available from the corresponding author upon reasonable request.

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
