# Peer review of "Accuracy and Safety Between Robot-Assisted and Conventional Freehand Fluoroscope-Assisted Placement of Pedicle Screws in Thoracolumbar Spine: Meta-Analysis"

_medicina, 2025, doi:10.3390/medicina61040690_

Round 1
Reviewer 1 Report
Comments and Suggestions for Authors
Excellent analysis of your study results. A key aspect of the article is the discussion of limitations, as different robotic systems offer varying capabilities. Additionally, the familiarity of spinal surgeons with these systems certainly influences the outcomes. Further studies are necessary to enhance the reliability of the results. At this stage, your approach is highly interesting, providing valuable insights to the literature.
Author Response
Thank you very much for your comments.
Kind regards
Reviewer 2 Report
Comments and Suggestions for Authors
Thank You for the opportunity to review this well written and thorough article! I would like to especially commend the authors for adhering to the PRISMA guidelines and the registration in PROSPERO!
The authors manuscript presents a comprehensive meta-analysis to evaluate the advantages or potential disadvantages of robot-assisted and conventional free-hand fluoroscope assisted placement of pedicle screws. The study synthesizes findings from the currently available evidence (which to be sure, could be more all-encompassing but that is not Your responsibility). The authors employed rigorous methodological frameworks and utilized appropriate statistical tools to draw their conclusions.
The meta-analysis demonstrates a high level of scientific rigor and soundness in several key areas.
The inclusion and exclusion criteria for study selection are well-defined and transparent. The authors' decision to focus on specific study types, e.g., only studies that include groups of 5 patients or larger is arbitrary though seems subjectively justified. The authors systematic approach to the literature search ensures that the selected studies are representative of the current body of research.
The methods used for data extraction are both thorough and consistent across the included studies. The authors clearly describe the process, which enhances the reproducibility of the study. The statistical techniques employed are appropriate for a meta-analysis. Heterogeneity analysis, and the assessment of publication bias are all conducted with proper statistical rigor.
Furthermore, the authors do an excellent job contextualizing their results within the broader literature. I only believe that a slightly more in-depth description of the individual underlying studies (statistical and methodological) limitations would be warranted. Especially considering the use of different robotic systems and maybe even different approaches to fluoroscopy-assisted techniques. This could fit well within the discussion section as the authors already mentioned this potential confounder there: “Furthermore, the use of different types of robotic systems and technologies across the studies adds complexity to the analysis, as each robot may have distinct capabilities and limitations that influence surgical performance and outcome…”. But The authors could expand on this a little bit by showcasing what systems were used and if differences in capabilities are already reported.
I do believe that the overall results still deliver valuable input despite the reported heterogeneity. But I would love to read the authors slightly more in-depth thought process on why the differences in surgical approaches and systems used may even be a benefit when considering the underlying research question.
Overall, this meta-analysis is an excellent contribution to the field, offering valuable insights grounded in rigorous methodology and careful statistical analysis. The scientific soundness of the study is evident, and it offers a comprehensive understanding of the topic. The few grammatical issues identified are minor and do not detract from the overall quality of the work. I would still love to see, if the authors could expand on their discussion section a bit further as I suggested – even though I would not consider the current state as a clear impasse on the articles route to publication.
After addressing these minor points, I deem the article ready for publication.
Author Response
Thank you for your comments and suggestions for improvement.
Thank you for pointing this out to us. The following changes were made based on the requested suggestions.
The typology of robotic systems is highly diverse. The robots utilized in this analysis are as follows: "TiRobot system" (TINAVI Medical Technologies Co. Ltd, Beijing, China), "SpineAssistTM" (Mazor Robotics, Caesarea, Israel), "Renaissance" (Mazor Robotics Ltd, Caesarea, Israel), "ExcelsiusGPS" (Globus Medical), “TianJi Robot System” and the "Orthbot system." Other major systems approved by the FDA include "ROSA" (Zimmer-Biomet), "Cirq" (Brainlab), "Curexo" (CUREXO Inc), "Kinguide" (Point Robotics), and "Remi" (Accelus). The cost of each platform ranges from $700,000 to $1.5 million. In a meta-analysis comparing the performance of the three most popular robotic platforms from 2016 to 2023, Maclean et al. reported excellent accuracy rates among the various robots (Mazor, Rosa, ExcelsiusGPS). On the other hand, they reported that both the ExcelsiusGPS and Mazor were associated with significantly reduced blood loss compared to ROSA. According to the data from our analysis, the accuracy appears to be high across all the described robots. Specifically, “TiRobot system” and “SpineAssistTM” appear to achieve optimal precision in terms of pedicle screw position accuracy. However, due to various confounders such as the heterogeneity of the underlying diseases and the number of patients involved, definitive conclusions regarding which robotic system is superior cannot yet be drawn.
For instance, the use of the robot may have a significant impact on cortical-bone trajectory (CBT) screws, as the CBT technique increases the contact surface area between the screws and cortical bone compared to the traditional method. According to the planned trajectory, the robotic arm moves to the targeted entry point. Once in position, the cannulas and dilators can be placed through the end effector. The use of robots may be particularly useful for CBT screws, as this technique follows a divergent trajectory, reducing the need for skeletonization. Therefore, the maneuvering angle of the robotic arm could facilitate this technique. In fact, in some cases, for classical pedicle screws using robots, it is necessary to create a trans-fascial cut through the same open wound or a separate percutaneous passage down to the bone. However, additional studies are required to determine the actual benefit of the robot in CBT spinal technique.
Reviewer 3 Report
Comments and Suggestions for Authors
Dear authors,
The manuscript for the meta-analysis review provides a comprehensive fusion of the literature on “Accuracy and safety between robot-assisted and conventional free-hand fluoroscope assisted placement of pedicle screws in thoracolumbar spine: a meta-analysis”. It is credible, given that the authors fully describe their methodology and criteria and emphasize the study's limitations and the need for more scientific studies to quantify robot-assisted surgery.
- The manuscript is well-organized and features a clear structure, including an introduction, methods, results, and discussion sections. The authors have referenced a diverse selection of articles and have applied appropriate criteria for their inclusion and exclusion. The statistical techniques used for the meta-analysis are robust and suitable for the study. The discussion thoughtfully interprets the findings, emphasizing both the strengths and limitations of the studies included.
Suggestions for Improvement:
- Graphical Abstract: While the manuscript is well-written, it would greatly benefit from the inclusion of a graphical abstract in a visually appealing format. A visual summary of the meta-analysis could enhance the reader's understanding of the study’s key findings and improve the manuscript's accessibility.
- The use of abbreviations and their expanded forms is inconsistent. Standardizing these will enhance the article's readability. For example, on Page 6, Line 20 “Wang and colleagues excluded significant differences between RS and fluoroscopy-assisted techniques”.
- Could you please clarify if any of the studies compared medically compromised patients with their length of hospitalization?
Author Response
Thank you for your comments and suggestions for improvement.
We agree that a visual summary could improve the manuscript's accessibility. We have prepared and included a graphical abstract in Supplementary Data.
We have corrected the use of abbreviations as fully as possible.
Some studies have compared medically compromised patients with their length of hospitalization, but only in a descriptive manner. A precise and comprehensive analysis of the influence of the robot in these patients has not been performed. This is an interesting point for future studies.